# Hydrogen Induced Etching Features of Wrinkled Graphene Domains

**DOI:** 10.3390/nano9070930

**Published:** 2019-06-28

**Authors:** Qiongyu Li, Fang Li, You Li, Yongping Du, Tien-Mo Shih, Erjun Kan

**Affiliations:** 1Department of Applied Physics and Institution of Energy and Microstructure, Nanjing University of Science and Technology, Nanjing 210094, China; 2Department of Mechanical Engineering, University of California, Berkeley, CA 94720, USA

**Keywords:** CVD synthesis, graphene, hydrogen etching, wrinkling

## Abstract

Wrinkles are observed commonly in CVD (chemical vapor deposition)-grown graphene on Cu and hydrogen etching is of significant interest to understand the growth details, as well as a practical tool for fabricating functional graphene nanostructures. Here, we demonstrate a special hydrogen etching phenomenon of wrinkled graphene domains. We investigated the wrinkling of graphene domains under fast cooling conditions and the results indicated that wrinkles in the monolayer area formed more easily compared to the multilayer area (≥two layers), and the boundary of the multilayer area tended to be a high density wrinkle zone in those graphene domains, with a small portion of multilayer area in the center. Due to the site-selective adsorption of atomic hydrogen on wrinkled regions, the boundary of the multilayer area became a new initial point for the etching process, aside from the domain edge and random defect sites, as reported before, leading to the separation of the monolayer and multilayer area over time. A schematic model was drawn to illustrate how the etching of wrinkled graphene was generated and propagated. This work may provide valuable guidance for the design and growth of nanostructures based on wrinkled graphene.

## 1. Introduction

As the first discovered two-dimensional system, graphene has already captured global attention owing to its extraordinary physical and chemical properties, which hold great potential for electronic and optoelectronic applications [1,2,3]. Compared to other synthesis methods, graphene grown on Cu (copper) by CVD has gained distinct advantages of having high quality wafer-scale single crystal films and being transferable to arbitrary substrates [4,5,6]. Studies have suggested that hydrogen plays a dual role as both the etching reagent and the growth activator when a hydrogen-diluted carbon source for CVD graphene growth is used [7,8,9]. Hydrogen etching is initiated by breaking carbon bonds and removing carbon atoms at locations with unstable energy, such as graphene edges or structure imperfection sites, and is proven to be a fascinating tool to fabricate nanoribbons, nanorings, shaped junctions, and other patterned graphene nanostructures with specific functions [10,11,12,13].

While it is a ubiquitous phenomenon for CVD graphene grown on Cu, the occurrence of wrinkles originates because of different thermal-expansion coefficients of graphene and Cu during post cooling down process, and can modify the electronic structure, transport properties, and surface properties of graphene [14,15,16,17,18,19]. Zhu et al. studied wrinkle topographies and structures by experimental and theoretical measurements and highlighted the coupling between morphology and electronic properties [17]. Liu et al. found that the distribution of wrinkles in CVD graphene depended on the Cu crystal structure, and the result indicated that the folded wrinkle tended to appear on Cu (001) facets, while the standing collapsed wrinkle appeared more easily on Cu (111) facets [20]. Even though wrinkling is a practical issue in CVD graphene, the hydrogen etching effect on wrinkled graphene has not been systematically studied, which is worthy of attention for the fabrication of high-quality, well-defined graphene nanostructures in various applications.

In this study, graphene domains with high density wrinkles were obtained by applying a fast cooling down process on polished Cu foil via CVD and then were etched by exposure to hydrogen/Ar flows at high temperatures. Significant differences in the wrinkle distribution and hydrogen-induced etching behaviors are observed for monolayer and multilayer graphene domains (multilayer graphene domains are specifically referred to as those graphene domains with a small portion of the multilayer area in the center in this work). By carrying out a systematic study along the time axis, a special etching pattern on wrinkled multilayer graphene domains is revealed and the corresponding mechanism is proposed. The result shown here may provide valuable guidance for the design and the growth of unique graphene nanostructures.

## 2. Experiments

A 25 μm thick Cu sheet (Stock No. 46365, Alfa Aesar Inc., Tewksbury, MA, USA) was electrochemically polished for graphene synthesis. After removing the polishing solution residuals, we folded the Cu sheet into a pocket-like structure in order to keep a clean and smooth surface [21,22]. The Cu pocket was loaded into a quartz tube from the low pressure chemical vapor deposition (LPCVD) system (Xiamen G-CVD) and heated to 1030 °C under an O_2_ flow of 0.5 sccm (5 mTorr). Then, a CH_4_ flow of 5 sccm (160 mTorr) and a H_2_ flow of 5 sccm (55 mTorr) were introduced to the CVD system for graphene growth, and the growth time was varied from 10 min to 60 min. After growth, the system was cooled down to room temperature with different cooling rates. We annealed the graphene samples in Ar/H_2_ gas mixture at 900 °C after growth to explore the influence of wrinkle-induced H_2_ etching behavior. Graphene formed on the inside surface of the Cu pocket was transferred onto 300 nm SiO_2_/Si substrates using the PMMA (Polymethyl Methacrylate)-assisted method for further characterization [23].

These prepared wrinkled graphene samples were analyzed by Optical microscopy, Raman, atomic force microscope (AFM), and scanning electron microscopy (SEM). Optical microscopy studies were carried out with the digital optical microscope (Metallurgical Microscope, MV5000, Nanjing Jiangnan Novel Optics Co., Ltd., Nanjing, China) in reflectance mode. The surface morphology of wrinkled graphene samples was characterized by SEM (Sigma, Zeiss Inc., Oberkochen, Germany) using an acceleration voltage of 10 kV. Finally, structure and bonding properties of the samples were characterized by Raman analysis using a Witec laser Raman spectrometer (Alpha-300, WITec Wissenschaftliche Instrumente und Technologie GmbH, D-89081 Ulm, Germany) with laser excitation energy of 488 nm, power of ~10 mW, and resolution of ~207 nm. Single spectra were acquired with 0.5 s integration time for 10 times accumulation. Mapping images were obtained for 10 s/line scan speed, 100 points per line, and 100 lines per image. The topography of wrinkled graphene samples was evaluated by AFM (Dimension Icon, Bruker, Karlsruhe, Germany).

## 3. Results and Discussion

The typical LPCVD parameter used for the wrinkled graphene synthesis is shown in Figure 1a. After growth, the Cu foil was fast cooled to room temperature within 5 min by directly moving the furnace away from the hot zone. On the electro-polished Cu foil, the monolayer graphene domain and multilayer graphene domain were obtained, as shown in Figure 1b,c, respectively. The inset of Figure 1c shows a higher resolution SEM image of the multilayer edge area as the white line squared in Figure 1b. The layer difference can be distinguished via brightness contrast and the darker area corresponds to the multilayer area [24]. Raman spectra taken from different areas are presented in Figure 1d. The red curve (taken from the region indicated by the red dot in Figure 1c) and the blue carve (taken from the region indicated by the blue dot in Figure 1b) have sharp G and 2D peaks, with a G/2D ratio of less than 0.5, which confirms that the located region is a monolayer. The green curve represents the Raman spectrum of the darker center area (as indicated by the green dot in Figure 1c). The intensity of the G peak is higher than that of the 2D peak, indicating the existence of a multilayer graphene [1,21]. This trend suggests that graphene nucleation and growth of individual domains appear inhomogeneous and multilayers can only contribute to a small portion of the surface, as the graphene growth is dominated by a self-terminated process on Cu [25,26]. Numerous white lines are observed across the entire surface of the monolayer graphene domain (Figure 1b). However, in the multilayer graphene domain, white lines are distributed only in the monolayer area and are all broken on the boundary of the multilayer graphene areas (Figure 1b). No white lines were found if a slow cooling environment was applied (Appendix A). Thus, the white line is bound up with the wrinkle formed during the fast cooling process [20]. 

More quantitative information about the formed wrinkle was obtained using AFM characterization. The AFM image of a representative wrinkled area that consists of a range of widths of graphene wrinkles from ~60 nm to ~100 nm is shown in Figure 2a. AFM heights of the thick, white line-marked region are shown in the inset and reach approximately 4 nm to 7 nm, respectively. The result obtained here agrees well with previous reports about the morphology of wrinkles formed in CVD graphene growth on Cu [17,18]. The interaction between the Cu substrate and graphene strongly influences wrinkle formation. During the fast cooling stage, residual strains will be induced through thermal quenching due to the opposite polarity of thermal expansion coefficients of graphene and Cu. By releasing the strain effect, CVD graphene tends to buckle up with the formation of wrinkles at weak interaction sites [16,18]. In comparison with monolayer graphene, more strain is needed to break the configuration of bilayer (or multilayer) graphene [27]. In this work, wrinkles tend to form in the monolayer area rather than the multilayer area, probably because the strain is too small here to break the configuration of multilayer area, as shown in Figure 2b. All wrinkled lines formed in the monolayer area are broken on the boundary of the multilayer area, resulting in a high density wrinkle breakpoint zone along the boundary (as seen in Figure 1c).

After the morphological characterization of wrinkled graphene, an H_2_ etching experiment was conducted at 950 °C under a H_2_/Ar mixture gas flow. Different etching features of wrinkled graphene domains are observed in the optical microscope image (Figure 3a). Figure 3b,c are SEM images of individual etched wrinkled monolayer and multilayer graphene domains, and hexagonal holes can be confirmed owing to the anisotropic etching process [7]. Our etching experiment was conducted immediately after the growth, and the sample was not taken out from the tube. We think there are few chances for the sample to absorb other foreign elements, except H_2_. Even though there may be a small amount of oxidant residuals (~0.1 sccm) in the local gas phase, they will be hydrogenated immediately in the highly reductive environment. The etching process is attributed to hydrogenation and volatilization of carbon atoms. Since dangling carbon atoms at graphene edges are more reactive, H_2_ molecules dissociate exothermally and form bonds with carbon atoms at edges first. The etching begins from the graphene edges by removing carbon atoms [28]. Meanwhile, as a major type of structural imperfection, wrinkles show preferential chemisorption to atomic hydrogen and enhanced reactivity to hydrogenation because of high curvature, leading to them being etched more easily than other graphene areas [29,30]. Figure 3d shows a magnified view of an etched area. Hexagonal holes tend to appear along the wrinkle lines. To give quantitative information of the etching preference on wrinkled sites, we further chose five different regions with a corresponding area of 100 μm × 120 μm from the sample for calculation of the etching sites on and off the wrinkles. On the basis of statistical results in Figure 3e, we calculated the mean values for the etched sites on and off wrinkles as 132 and 7, respectively. The percentage of the etched sites on and off wrinkles reaches up to 18.9%.

In the monolayer graphene domain, the etching initiated from the edge and the wrinkled sites (Figure 3b). Now, we observe an interesting etching nanostructure in the wrinkled multilayer graphene domain, as clearly shown in Figure 3c. As the boundary of the multilayer graphene serves as a high density wrinkle zone, the etching also starts from the multilayer boundary, which results in the separation of multilayer and monolayer areas. Raman characterization was also used to identify the structure of the graphene after etching. Raman spectra were taken from the remaining monolayer graphene area, the flower-like multilayer area in the center, and one of the etched hexagons, respectively. The red curve in Figure 3f represents the Raman spectrum of the remaining monolayer graphene (pointed by the red arrow in Figure 3d), which does not show the presence of a D peak, indicating the remaining graphene after etching has few defects [31]. The blue curve represents the Raman spectrum inside the etched hexagons (pointed by the blue arrow in Figure 3d), which almost does not show any G and 2D band intensities, indicating the removal of graphene by anisotropic etching of hydrogen. The green curve represents the Raman spectrum of the remaining center area of the multilayer graphene domain, as pointed out by the green arrow in Figure 3c. The intensity of the G peak is higher than that of the 2D peak, indicating the existence of more than a single graphene layer. 

To further confirm the etching feature, we transferred the etched multilayer graphene domain onto the 300 nm SiO_2_/Si wafer and conducted Raman studies to investigate the structure properties. Figure 4 represents optical images with corresponding Raman spectra and maps of the G and 2D bands. The morphology of etched graphene domain remains undamaged after the transfer process and the thickness, and the uniformity of the etched wrinkled graphene domain can be evaluated via color contrast under an optical microscope (Figure 4a). In addition to the many small anisotropically-etched, hexagonal holes in the monolayer area, a larger hole with regular edges and an angle of 120 degrees along the multilayer area can be also obtained. Figure 4b,c shows Raman maps of the G band (1510–1650 cm^−1^) and 2D band (2650–2750 cm^−1^) with different brightness contrasts, indicating the occurrence of monolayer, bilayer, and trilayer graphene. In wrinkled regions, there are peak height variations in both the G and 2D bands [17], and there is a broadening of the 2D band that results in the darker line in the monolayer area, as clearly shown in the G map. Raman spectra (Figure 4d) are taken from the spots marked with the corresponding colored arrows shown in Figure 4a. The black curve represents the Raman spectrum inside one of the etched hexagons (pointed by the black arrow), which only shows a signal of Si, indicating no graphene residual after hydrogen etching. The blue curve in the inset of Figure 4d represents the Raman spectrum of the remaining monolayer graphene (pointed by the blue arrow), with ~0.5 G/2D intensity ratio and ~36 cm^−1^ full width at half maximum (FWHM) of the 2D peaks. The red curve corresponds to the Raman spectrum of the bilayer graphene area, as shown by the red arrow in Figure 4a, which presents ~1.5 I_G_/I_2D_ and a FWHM of 2D band of ~52 cm^−1^, suggesting the strong interlayer coupling of the bilayer graphene [31,32].

To better understand the etching reaction of wrinkled multilayer graphene, we investigated the influence of time on the etching process carefully. A typical optical microscopy image of the as-grown multilayer graphene domain transferred to 300 nm SiO_2_/Si is shown in Figure 5a, where the hexagonal shape of the multilayer area can be observed clearly in the center of the graphene domain via the contrast. Figure 5b–d is the optical microscopy image of the wrinkled graphene domain after etching for 10 min, 20 min, and 30 min, respectively. During the etching process, in addition to the numerous small hexagonal holes generated in the monolayer area, a larger hexagonal hole around the multilayer area appeared, as highlighted in the blue hexagonal frame. An important point is that the edge of the large hexagonal hole and the multilayer graphene appear parallel, indicating that the crystal orientation of the etching process resembles the growth process [6,13]. As the etching time increases, the large hexagonal hole extends from two directions by removing C atoms from the single layer area and multilayer area simultaneously. To achieve a quantitative understanding of the etching process, we studied the mean value of the multilayer area and the size evolution of the hexagonal hole along the multilayer as a function of etching time, with results shown in Figure 5e. The original value of the as grown multilayer area is ~72 μm^2^ and was reduced to ~48 μm^2^ and ~12 μm^2^ after etching for 10 min and 20 min, respectively, giving an etching rate of ~3 μm^2^/min of multilayer graphene. When the etching time increased to 30 min, the multilayer area became invisible. While the diameter of the hexagonal hole along the multilayer area in the center region was ~18 μm, ~39 μm, and ~63 μm after etching for 10 min, 20 min, and 30 min, respectively, indicating that the broadening of the hexagonal hole accelerated with a nearly linear behavior. We also observed a considerably faster etching rate of ~133 μm^2^/min for monolayer graphene (Appendix A). Therefore, it can be concluded that the hexagonal hole appearing around the multilayer area can be enlarged effectively by removing carbon atoms from two opposite directions: the interior edge of the monolayer and edges of the multilayer graphene.

Based on the finding of our experiments and facts from previous reports [7,8,10], a possible schematic diagram for the etching process of the wrinkled multilayer graphene domain was drawn in Figure 6. In the fast cooling process, wrinkles tend to appear on the monolayer area and break on the boundary of the multilayer area, which results in a high density structure imperfection zone around the multilayer graphene area. Furthermore, hydrogen atoms, absorbed on the wrinkled graphene surface, have two main preference sites: edges and wrinkle regions. Therefore, we propose that when annealing wrinkled multilayer graphene in H_2_/Ar gas, active hydrogen species attach to the exterior edge of the monolayer graphene and wrinkled sites first and start to remove carbon atoms, creating small anisotropically-etched hexagonal holes along edges and wrinkle lines within monolayer graphene areas. As the boundary of the multilayer acts as a high density wrinkle zone, a larger hexagonal hole with parallel edges to the multilayer will form over time, ending up with the separation of the monolayer and multilayer areas. Afterwards, there are four dominant processes for removing carbon atoms on the Cu surface that contribute to the specific etching feature: (1) the removal of dangling carbon atoms from exterior edges of monolayer graphene; (2) etching from interior edges of monolayer graphene; (3) taking carbon atoms away from the edge of the multilayer graphene; (4) hydrogenation at other wrinkled sites. 

## 4. Conclusions

We have demonstrated layer-dependent wrinkle distributions of graphene grown on Cu by LPCVD and the special hydrogen-induced etching phenomenon of wrinkled multilayer graphene domains. Under a relatively low strain condition, wrinkles tend to form in the monolayer area and break on the boundary of the multilayer area, resulting in a high density wrinkle zone along the multilayer area in the multilayer graphene domains. Due to the preference absorption of hydrogen atoms on wrinkled sites, the etching initiated from the boundary of the multilayer area, creating a large hexagonal hole to separate the monolayer and multilayer area. Then, the etching of the wrinkled multilayer graphene domain is accelerated through four feasible paths, including the exterior and interior edge of the monolayer graphene, edges of the multilayer graphene, and wrinkled sites. These findings shed light on the formation of wrinkles in CVD growth graphene domains, as well as their selective etching process, to fabricate well-defined structures.

## Figures and Tables

**Figure 1 nanomaterials-09-00930-f001:**
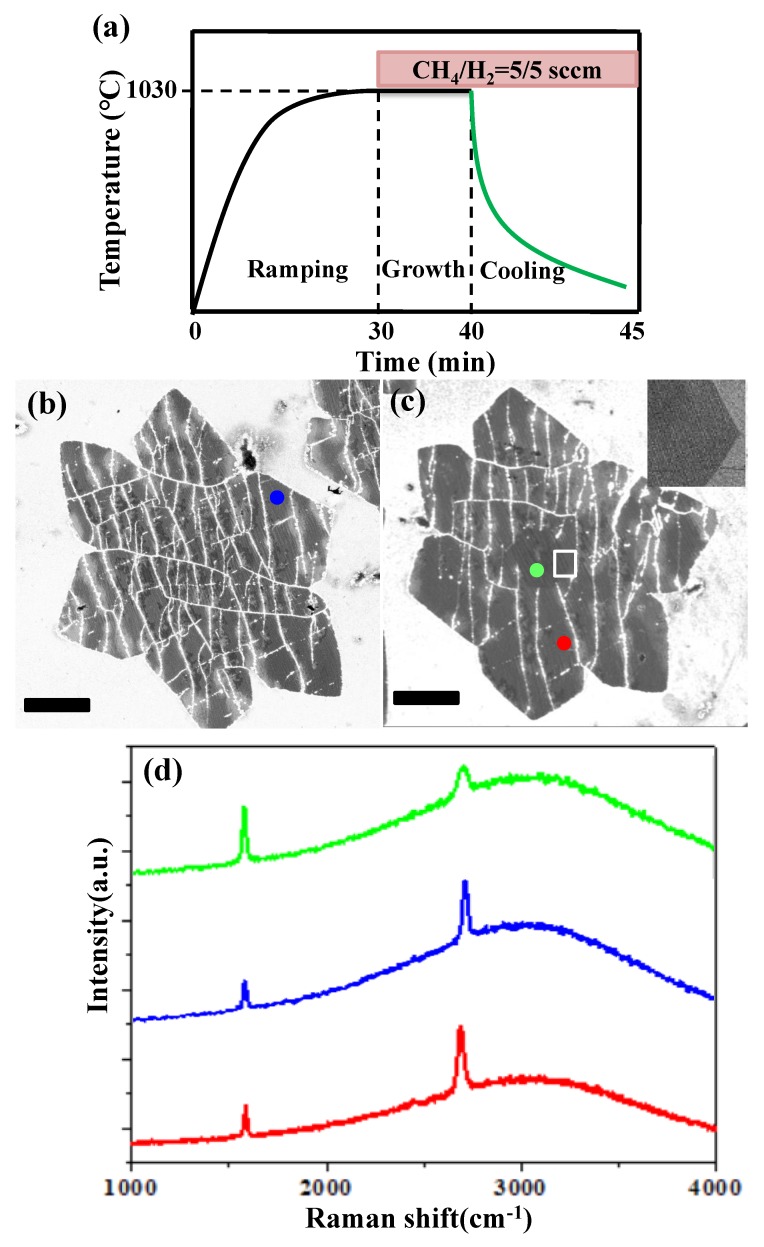
(**a**) LPCVD synthesis parameters used for the wrinkled graphene synthesis. (**b**) SEM image of as synthesized wrinkled monolayer graphene domain. (**c**) SEM image of as synthesized wrinkled multilayer graphene domain and inset is the magnified picture of the multilayer edge. (**d**) Raman spectra of different graphene areas shown by the colored dot in (**b**, **c**). All scale bars are 5 μm.

**Figure 2 nanomaterials-09-00930-f002:**
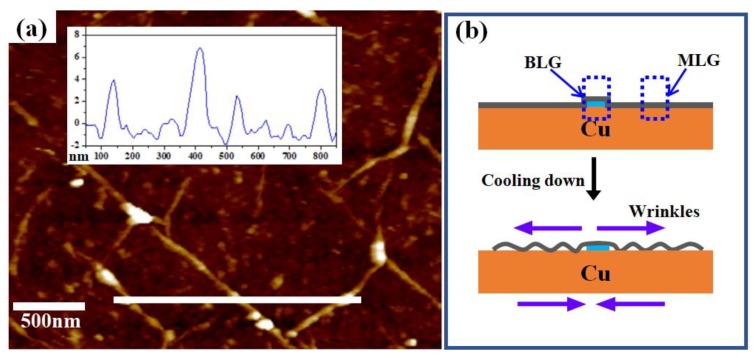
(**a**) AFM image of wrinkled graphene transferred onto SiO_2_/Si and the inset is the height of the thick, white line-marked region. (**b**) Schematic diagrams showing the effect of thermal expansion mismatch on the formation of wrinkles in monolayer graphene (MLG) and bilayer graphene (BLG) areas.

**Figure 3 nanomaterials-09-00930-f003:**
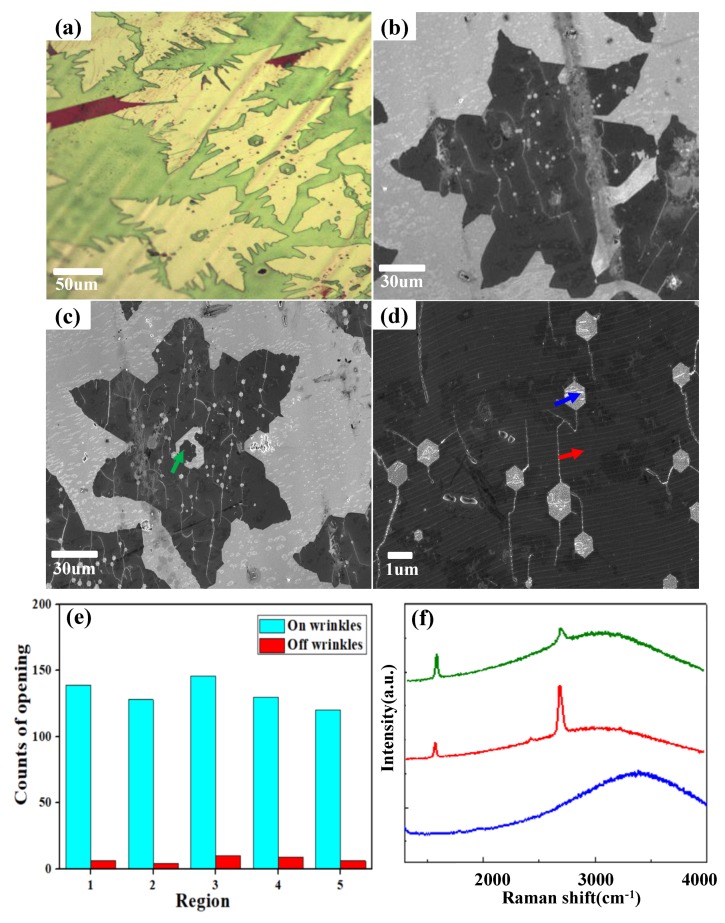
(**a**) Optical microscope image of H_2_-induced etching of wrinkled graphene domains. (**b**) SEM image of the etched monolayer wrinkled graphene domain. (**c**) SEM image of the etched multilayer wrinkled graphene domain. (**d**) Magnified SEM image of etching features of wrinkled graphene. (**e**) Percentage of etched openings on and off wrinkles. Five regions were randomly picked for the calculation of etched openings; each region is 100 μm × 120 μm. (**f**) Raman spectra of the intact monolayer graphene (pointed by red arrow), etched region (pointed by blue arrow), and multilayer graphene (pointed by green arrow) on Cu foil.

**Figure 4 nanomaterials-09-00930-f004:**
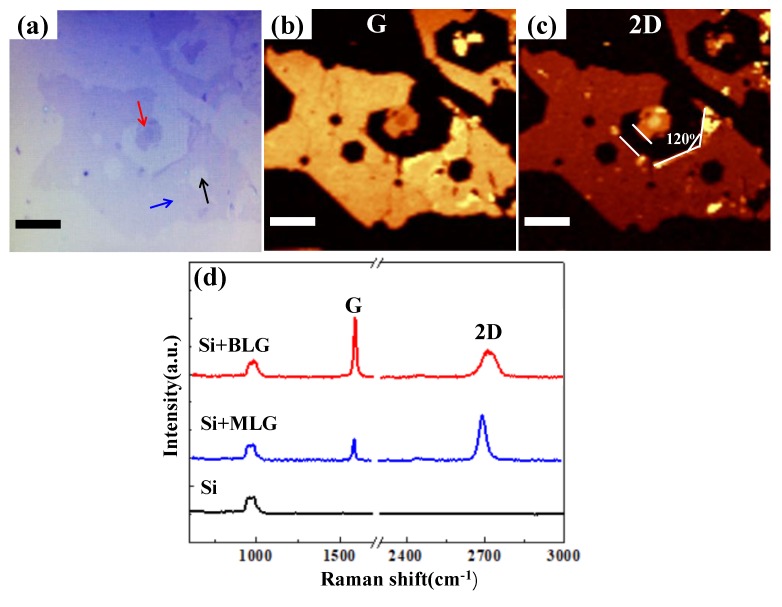
(**a**) Optical microscope image of the etched graphene domain after transferring to SiO_2_/Si substrate. (**b**,**c**) Raman G band map (1510–1650 cm^−1^) and 2D band map (2650–2750 cm^−1^). (**d**) Typical Raman spectra taken from the monolayer graphene (MLG) area, bilayer graphene (BLG) area, and etched hexagonal holes (**a**). All scale bars are 20 μm.

**Figure 5 nanomaterials-09-00930-f005:**
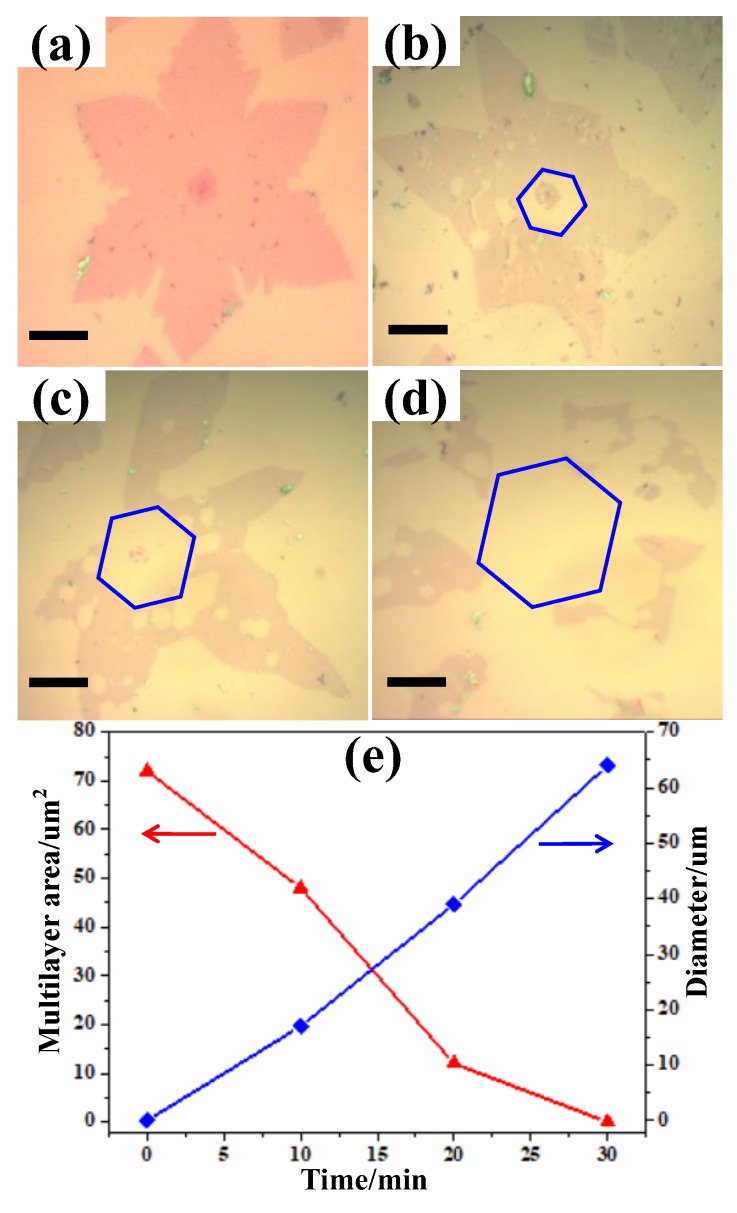
Optical microscope images of as-grown and etched wrinkled multilayer graphene domains transferred to the SiO2/Si substrate. (**a**) The original grown wrinkled multilayer graphene. (**b**–**d**) The etched graphene obtained after etching for (**b**) 10 min, (**c**) 20 min, and (**d**) 30 min. (**e**) Statistical data of the area of the multilayer and the diameter of the hexagonal hole along the multilayer area with etching time. All scale bars are 20 μm.

**Figure 6 nanomaterials-09-00930-f006:**
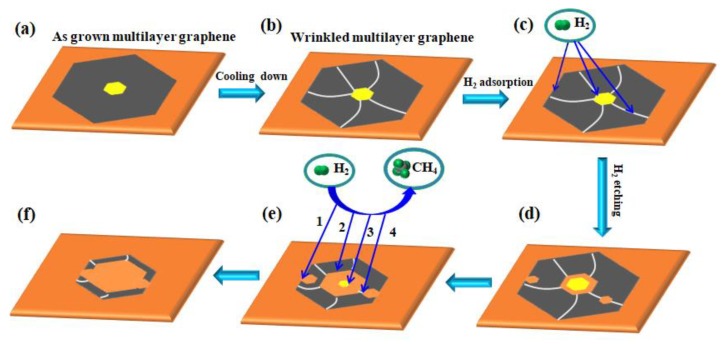
The schematic diagram for the growth and etching of the wrinkled multilayer graphene domain. (**a**) The multilayer graphene domain was grown on a polished Cu surface. (**b**) Formation of wrinkles under fast cooling condition. (**c**) H_2_ chemisorbed on the energy unstable sites. (**d**–**f**) Etching generated and propagated through four pathways: (1) exterior edges of the monolayer graphene; (2) interior edges of the monolayer graphene; (3) edges of the multilayer graphene; (4) wrinkled sites.

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
