# Peer review of "Hydrogen Induced Etching Features of Wrinkled Graphene Domains"

_nanomaterials, 2019, doi:10.3390/nano9070930_

Round 1
Reviewer 1 Report
The manuscript by Li et al. reports on the effect of hydrogen etching on wrinkled graphene areas on copper.
I have some concerns that I ask the authors to address:
- Figure 1 displays two SEM images of what the authors call monolayer (Figure 1b) and multilayer (Figure 1c) domains. However, I do not find enough evidence of those domains to be mono- or multi-layer. The authors speak of different contrast in the SEM images, but the contrast can be due to several different features, like for instance different orientation of the Cu grains. Thus, I would require to carry out Raman spectroscopy in order to solve this issue.
- The authors explain the H2 etching process saying that H2 molecules dissociate and then react with the C atoms at the edge of the graphene domains. This is reasonable, however, I would appreciate if the authors could elaborate more on the H2 dissociation step. Where does it happen? Still concerning this issue, the authors should try an experiment where they try and etch a sample of copper fully covered by graphene (not a sample made of isolated domains as the current samples).
- Minor comment: when possible, the figures should be made larger to make it easier for the reader to pinpoint details in them.
Author Response
We thank the referee for considering our work especially timely, carefully reading the manuscript and the valuable suggestions. We answer all of the referees’ comments in detail, and describe the changes made.Please find the attachment.

Reviewer 2 Report
The manuscript is quite well written and original.
Author Response
We thank the referee for considering our work especially timely, carefully reading the manuscript.
Reviewer 3 Report
Please find the full comments attached as a separate word document.

Author Response

(The authors gave the same response as above.)

Round 2
Reviewer 1 Report
The authors have addressed the points I addressed in my previous report.
In the methods section, details of the parameters used for performing Raman spectroscopy and scanning electron microscopy need to be included in the text.
Author Response
We thank the referee for this comment and we have addressed the point by point response as below.
Point 1: In the methods section, details of the parameters used for performing Raman spectroscopy and scanning electron microscopy need to be included in the text.
Response 1: We appreciate the thorough review from the referee. We have made corresponding revisions as list of modification 1.
Modifications:
1. Line 74
The surface morphology of wrinkled graphene samples was characterized by SEM (Zeiss Sigma). Finally, structure and bonding properties of the samples were characterized by Raman analysis using Witec laser Raman spectrometer with laser excitation energy of 488 nm, with the topography evaluated by AFM (Dimension Icon).à The surface morphology of wrinkled graphene samples was characterized by SEM (Zeiss Sigma) using an acceleration voltage of 10 kV. Finally, structure and bonding properties of the samples were characterized by Raman analysis using Witec laser Raman spectrometer (Alpha-300) with laser excitation energy of 488 nm, power of ~10 mW and resolution of ~207 nm. Single spectra were acquired with 0.5s integration time for 10 times accumulation. Mapping images were obtained for 10 s/line scan speed, 100 points per line, and 100 lines per image. The topography of wrinkled graphene samples was evaluated by AFM (Dimension Icon).